# AS-LLM: WHEN ALGORITHM SELECTION MEETS LARGE LANGUAGE MODEL

## ABSTRACT

Algorithm selection aims to identify the most suitable algorithm for solving a specific problem before execution, which has become a critical process of the AutoML. Current mainstream algorithm selection techniques rely heavily on feature representations of various problems and employ the performance of each algorithm as supervised information. However, there is a significant research gap concerning the consideration of algorithm features. This gap is primarily attributed to the inherent complexity of algorithms, making it particularly challenging to find a universally effective feature extraction method that is applicable across a diverse range of algorithms. Unfortunately, neglecting this aspect undoubtedly impacts the accuracy of algorithm selection and indirectly necessitates an increased volume of problem data for training purposes. This paper takes a significant stride towards addressing this gap by proposing an approach that integrates algorithm representation into the algorithm selection process. Specifically, our proposed model employs distinct modules to extract representations of both problems and algorithms, where the algorithm representation leverages the capabilities of pre-trained LLMs in the realm of code comprehension. Following the extraction of embedding vectors for both algorithms and problems, the most suitable algorithm is determined through calculations of matching degrees. Our experiments not only validate the effectiveness of the proposed model but also showcase the performance of different embedded pre-trained LLMs, which suggests that the proposed algorithm selection framework holds the potential to serve as a baseline task for evaluating the code representation capabilities of LLMs. The code will make publicly available after the review process.

## 1 INTRODUCTION

Performance complementarity, a phenomenon where no single algorithm consistently outperforms all others across diverse problem instances, is a well-established reality in the realm of optimization and learning problems (Kerschke et al., 2019). Over the past few years, the growing interest in automated algorithm selection techniques has become evident. These techniques aim to tackle the challenge of selecting the most appropriate algorithm from a predefined set for a given problem instance automatically (Ruhkopf et al., 2022; Heins et al., 2023). As depicted in Figure 1(a), existing techniques predominantly rely on two sources of information: (1) the features of each problem instance and (2) the historical performance of various algorithms across problem instances (Pio et al., 2023). Machine learning methods are then employed to establish a mapping from problem features to a subset of algorithms that yield optimal performance. Consequently, extensive research has focused on two critical aspects within this field: (1) designing problem feature extraction methods tailored to specific problem categories or tasks (Alissa et al., 2023), and (2) constructing advanced machine learning models to map problem features to algorithms (Tornede et al., 2022).

However, it is noteworthy that there has been a conspicuous absence of research focusing on the features of the algorithms themselves. Most current research has centered on problem features, treating algorithm-related information merely as a supervisor. For instance, some studies treat the selected algorithms as labels, modeling the task as either single-label (Brazdil & Giraud-Carrier, 2018) or multi-label (Dantas & Pozo, 2020) classification. Recognizing the inherent complexity and diversity of algorithms, quantifying and describing their features can indeed be a formidable challenge, and a universal representation method applicable across different algorithms remains

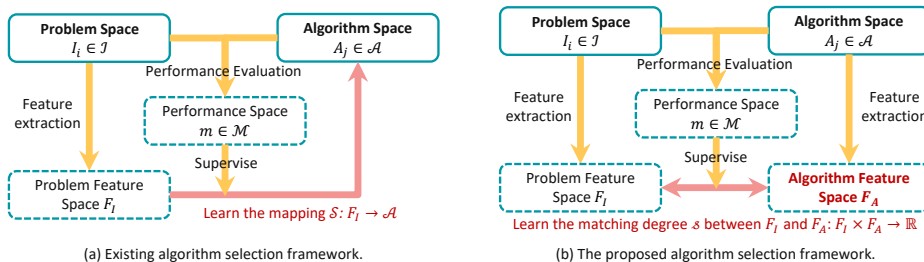

Figure 1: Comparison of the existing framework and the proposed framework.

elusive. Nevertheless, neglecting this critical aspect of algorithm features undeniably affects the overall model performance, posing at least three issues. Firstly, disregarding algorithm features as an essential information source inevitably results in a loss of model accuracy. Furthermore, relying solely on problem features often implies a unidirectional relationship, characterized by a one-way mapping from problems to algorithms. This unidirectional mapping does not align with the underlying bidirectional nature of the relationship between algorithms and problems, potentially missing crucial information that could enhance the model's performance. Additionally, neglecting algorithm features could potentially slow down the convergence, necessitating larger training data. This contradicts the essence of algorithm selection, which seeks to reduce experimentation costs as a preprocessing step. Requiring substantial and hard-to-acquire training data, such as performance data across diverse problems, undermines the original intent of algorithm selection.

On the other hand, the algorithm set is typically much smaller than the problem set, and the candidate algorithm set usually remains fixed during the training process (Cunha et al., 2018). Therefore, once algorithm features can be effectively extracted, they offer a convenient and potent resource. With the advent of the era of the pre-trained large language model (LLM) (Ouyang et al., 2022), extracting algorithm features has become more achievable. Code text data accurately represents the functionality and characteristics of an algorithm, and, with the assistance of pre-trained LLMs or dedicated pre-trained models for code text (Chen et al., 2021), we can represent code features with minimal training overhead. The universality of this extraction process may even surpass that of problem feature extraction.

Specifically, this paper introduces a novel algorithm selection framework, as depicted in Figure 1(b), which not only leverages information about problem features and algorithm performance but also captures algorithm representations. Accordingly, the learning objective of our model diverges from existing techniques. Instead of the one-way mapping from problems to algorithms, we directly model the matching degree between algorithm representations and problem representations. To achieve this goal, we propose an Algorithm Selection Model based on Large Language Model (AS-LLM) and deploy it in heuristic algorithm selection tasks for continuous optimization problems to demonstrate the merits of considering algorithm features. The AS-LLM model comprises two distinct tracks for extracting features from problems and algorithms. Problem features are traversed from the tree representation of the objective function, while algorithm features are extracted from corresponding code text. After passing through multiple layers of networks, the similarity computation between algorithm and problem representations determines the selected algorithm. The key contributions are summarized as follows:

- To the best of our knowledge, this paper pioneers the integration of algorithm features into the algorithm selection process, utilizing pre-trained LLMs for extracting features from algorithm code text. This method holds generality across various algorithms.

- The proposed AS-LLM offers at least three practical benefits: (i) A more accurate modeling of the bidirectional nature of algorithm selection tasks; (ii) Superior performance with fewer training samples; (iii) Versatility to adapt to various problem types simply by modifying the track used for problem feature extraction.

- Within the AS-LLM framework, the algorithm selection task serves as a valuable benchmark for evaluating the algorithm comprehension capabilities of LLMs. We integrate various LLMs into AS-LLM and empirically demonstrate their performance differences.

## 2 BACKGROUND

Algorithm selection, first proposed by Rice (1976), aims to choose the appropriate algorithm for each problem instance from a set of algorithms. Traditionally, the problem of per-instance algorithm selection can be defined as follows: Given an instance set $\mathcal{I}$ of problem set $P$, a algorithm set $\mathcal{A}$ for solving $P$, and a performance metric $\mathcal{M}: \mathcal{I} \times \mathcal{A} \to \mathbb{R}$ which quantifies the performance of any algorithm $A_j \in \mathcal{A}$ on each instance $I_i \in \mathcal{I}$, per-instance algorithm selection should construct a selector $\mathcal{S}: \mathcal{I} \to \mathcal{A}$ that assigns any problem instance $I_i \in \mathcal{I}$ to an algorithm $A_j \in \mathcal{A}$, optimizing the overall performance expectation $\mathbb{E}[\mathcal{M}(I_i, \mathcal{S}(I_i))]$ on $\mathcal{I}$ according to the metric $\mathcal{M}$ (Kerschke et al., 2019).

As outlined in the aforementioned problem definition, it is evident that algorithm selection is fundamentally straightforward. It involves the design of a machine learning model describing $\mathcal{S}$, where problem instances and their corresponding features serve as training samples. When faced with a new problem, the process entails feature extraction from the problem instance and subsequent input into the model to determine the selected algorithm. This approach aligns with the prevailing methodologies found in the state-of-the-art literature, which is reviewed as follows. We will delve into the two fundamental stages of existing methodologies individually: the extraction of problem features and the construction of algorithm selection model.

Feature Extraction for Problem Instances: Since existing techniques exclusively rely on problem features for modeling, the quality of these features significantly impact the performance of the algorithm selection process. Consequently, substantial research efforts have been dedicated to the extraction of problem features. The nature of the problem at hand dictates the selection of specific feature sets, as comprehensively summarized in literature (Kerschke et al., 2019). Typical features encompass statistical attributes derived from problem instances, e.g., the number of objectives and variables in optimization problems, as well as more sophisticated features like attributes of network graph (Zhu et al., 2018) or decision tree (An & Zhou, 2022) derived from each problems. Notably, certain research undertakes the extraction of performance-related information through brief algorithm executions for given problem instances (Maldonado et al., 2023), e.g., feature matrix formed by the information collected in the evolutionary process (Qiao et al., 2022). Although these features entail higher acquisition costs, they often yield superior performance outcomes (Hutter et al., 2014).

Research on Algorithm Selection Models: Existing methodologies leverage diverse learning models to establish the mapping $\mathcal{S}$. Some approaches employ classification models, treating each algorithm as a distinct category for algorithm selection (Wang et al., 2014; Tian et al., 2020). Alternatively, regression models are utilized to predict the performance of each algorithm, and the algorithm with the best predicted performance is chosen (Jankovic et al., 2021; de la Rosa-Rivera et al., 2021). Furthermore, certain methods employ clustering techniques to partition problem instances into distinct clusters within the feature space, enabling separate decisions for each cluster (Kadioglu et al., 2010; Song et al., 2023) with various decision techniques like hierarchical models. Additionally, some studies transform the algorithm selection procedure into a recommendation task (Mısır & Sebag, 2017; Yang et al., 2019a; Zhao et al., 2021). These approaches employ collaborative filtering techniques or similar methods to provide algorithm selection.

## 3 ALGORITHM SELECTION BASED ON LLMS

This section provides an overview of the main details of AS-LLM and conducts a theoretical analysis about the training method. In line with the definition of algorithm selection outlined in Section 2, research on algorithm selection conventionally entails defining a specific problem space $P$ and a candidate algorithm set $\mathcal{A}$. Throughout the remainder of this paper, our primary focus will be on the selection of meta-heuristic algorithms in the context of continuous optimization problems. It's important to highlight that the delineation of the problem space does not restrict AS-LLM's applicability solely to this specific scenario, and we will elaborate on its broader potential in the subsequent discussion of the algorithm. As illustrated in Figure 2, AS-LLM comprises three core components: problem feature extraction, algorithm feature extraction, and the subsequent calculation of similarity between these features. We will now introduce each of these components in detail.

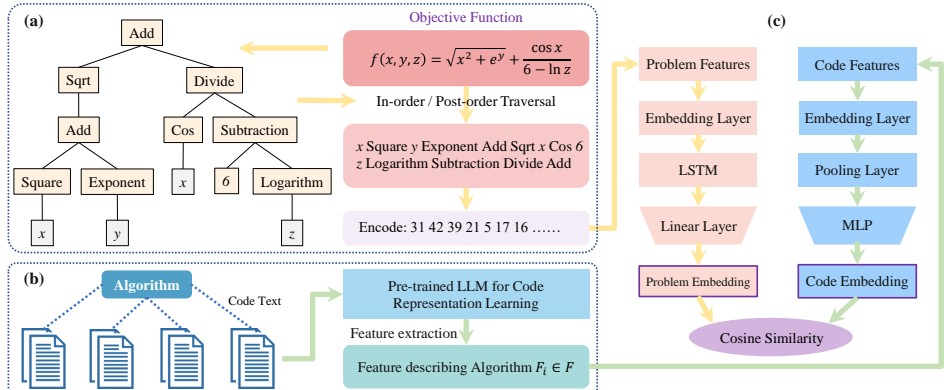

Figure 2: Illustration of AS-LLM: (a) An example of the feature extraction for problems. (b) The feature extraction method for algorithms. (c) The calculation of the matching degree between problems and algorithms.

## 3.1 PROBLEM REPRESENTATION

In the context of continuous optimization problems, capturing features of the objective function is a crucial step. One innovative approach to achieve this involves representing the objective function as a tree structure and performing a traversal of this tree, as shown in Figure 2(a). This method offers a unique perspective on automatic feature extraction of objective functions. By structuring the objective function as a tree, we can dissect its underlying components and relationships between variables and operations. As for expression traversal methods, it is essential to note that the in-order traversal typically results in the generation of the original infix expression, which often requires consideration of operator precedence and parentheses. On the other hand, the post-order traversal is employed to transform an expression into a postfix notation, commonly known as Reverse Polish Notation (Krtolica & Stanimirović, 1999), making it more suitable for computational evaluation by computers, which allow us to systematically explore the function's structure, preserving not only the mathematical expressions but also the hierarchical information within the function. Subsequently, the strings obtained during the traversal are subjected to embedding techniques, such as Word2Vec (Mikolov et al., 2013) or FastText (Bojanowski et al., 2017), to transform these strings into numerical vectors that encapsulate semantic meaning and relationships between different components of the objective function.

## 3.2 ALGORITHM REPRESENTATION

Algorithm features play a pivotal role in the emphasis of this paper. As shown in Figure 2(b), for each distinct algorithm under consideration, the corresponding code snippets in different programming languages is a readily accessible resource. Leveraging the code understanding capabilities of pre-trained LLMs, we aim to extract and characterize the essence of each algorithm from these code texts. This encompasses not only the original code used to generate performance space $\mathcal{M}$ but also algorithm functions with the same logic or functionality and even pseudocode representations. In this endeavor, the LLM employed can take on multiple forms, as long as its pre-training data contains code text. Hence, it can either be a general-purpose model pre-trained on a vast corpus of text data (e.g., GPT-3 (Floridi & Chiriatti, 2020)) or a specialized model tailored for code comprehension and generation (e.g., CodeBERT (Feng et al., 2020)). The model's ability to comprehend code syntax, structure, and semantics allows it to identify critical algorithmic patterns and techniques. This entails recognizing loops, conditionals, data structures, and their interplay within the code.

## 3.3 MODEL ARCHITECTURE

Our proposed AS-LLM is designed to provide accurate match by simultaneously extracting features from two distinct tracks: the problem track and the algorithm track, as shown in Figure 2(c). The

model computes the cosine similarity between the features extracted from these tracks and leverages this similarity score to select the most suitable algorithm for each given problem.

**Problem Track**: The problem track is responsible for capturing essential features from the traversal sequence of problem, obtained in Section 3.1. The input, represented as a sequence of tokens $\{q_1, q_2, \ldots, q_m\}$, is initially transformed into dense vectors using an embedding layer. This layer maps each token to a fixed-size continuous vector, and calculated as:

$$\mathbf{E}_P = \text{Embedding}((q_1, q_2, \ldots, q_m)) \in \mathbb{R}^{m \times s}, \tag{1}$$

where the dimensionality of these embeddings is determined by a hyperparameter $s$. Following the embedding layer, the model employs an Long Short-Term Memory networks (LSTM) based encoder. The encoder utilizes the embedded problem vectors and captures the order of variables and operators within the objective function. The LSTM encoder processes the embedded question matrix $\mathbf{E}_P$ to obtain the hidden states $\mathbf{H}$ and cell states $\mathbf{C}$ as follows:

$$\mathbf{H}, \mathbf{C} = \text{LSTM}(\mathbf{E}_P) \tag{2}$$

The LSTM encoder's final hidden state is concatenated with the initial hidden state and passed through a linear layer to fix the dimensionality as $n$:

$$\mathbf{F}_p = \text{Linear}(\text{Concatenate}(\mathbf{h}_f, \mathbf{h}_0)) \in \mathbb{R}^n \tag{3}$$

where $\mathbf{h}_f, \mathbf{h}_0 \in \mathbf{H}$. The vector $\mathbf{F}_p$, referred to as the problem representation vector, encapsulates the essential characteristics of the input problem.

**Algorithm Track**: The algorithm track focuses on extracting features related to selected algorithms. Algorithm indices are mapped to corresponding continuous vectors, obtained from Section 3.2 using a pre-trained LLM:

$$\mathbf{E}_A = \text{LLM-Embedding}(j) \in \mathbb{R}^e \tag{4}$$

where $j$ is the index of the algorithm, and $e$ is determined by the output scale of the chosen LLM. These embeddings can be frozen during training to preserve the knowledge encoded within them. When there is enough training data available, parameters in the LLM can be fine-turned together with the other parts of the model and then update the embeddings. Due to the potentially vast magnitude of $e$, which can range in the hundreds of thousands or even millions, there exists a significant disparity in dimensionality between algorithm features and problem features. Consequently, algorithm features must undergo pooling layer processing to mitigate their excessive influence on the final outcomes. A one-dimensional average pooling operation is applied to $\mathbf{E}_A$:

$$\mathbf{I} = \text{Pooling}(\mathbf{E}_A). \tag{5}$$

The vector $\mathbf{I}$ is then passed through an two-layer MLP with ReLU activation functions:

$$\mathbf{F}_A = \text{Linear}(\text{ReLU}(\text{Linear}(\mathbf{I}))) \in \mathbb{R}^n, \tag{6}$$

which transforms the algorithm representation $\mathbf{I}$ into $n$-dimensional vector $\mathbf{F}_A$, same size as the $\mathbf{F}_P$.

**Algorithm Selection**: After extracting features from both tracks, the model computes the cosine similarity between the problem vector and the algorithm vector, which can be calculated as:

$$\text{Similarity}(\mathbf{F}_A, \mathbf{F}_P) = \frac{\mathbf{F}_A \cdot \mathbf{F}_P}{\|\mathbf{F}_A\| \cdot \|\mathbf{F}_P\|}. \tag{7}$$

Cosine similarity measures the cosine of the angle between these vectors and provides a score indicating their similarity. A higher similarity score suggests a better match between the user's question and the algorithm. Finally, the similarity is concatenated with problem features and algorithm features, and jointly influence the model's final output through a MLP layer.

Traditional algorithm selection relies solely on problem features, allowing for direct use of the corresponding algorithm indexes or performance as labels, making the training process relatively straightforward. However, when incorporating algorithm features into AS-LLM, a reconstruction of training samples becomes necessary, with a critical focus on sampling of negative instances. The input of AS-LLM is problem-algorithm pair, where a positive sample signifies that the algorithm is the best-performed algorithm for the problem, while conversely, it constitutes a negative sample. To train AS-LLM, each training iteration involves a single sample, requiring the model to predict the positivity (matched) or negativity (unmatched) of each problem-algorithm pair. Similarity values are passed through a sigmoid function, and cross-entropy is employed as the loss function.

### 3.4 THEORETICAL ANALYSES: HOW TO CHOOSE NEGATIVE SAMPLES?

In this subsection, we try to analyse how to choose unselected algorithms (negative samples) during the training process. Algorithm selection problems typically prioritize the accuracy of the top-$k$ algorithms in the results. Hence, in the following theoretical analyses, we employ the Partial Area Under the Curve (Partial AUC) as a measurement. Partial AUC is particularly well-suited for evaluating the ranking quality of algorithms in situations where only the highest-ranked selections are of primary interest, which is formulated as:

$$\widehat{OPAUC}(\beta) = \frac{1}{|P|} \sum_{p \in P} \frac{1}{|A - \{A_p\}|} \sum_{i \in [\![A - \{A_p\}]\!]_\beta} \mathbb{I}\left(\text{Similarity}(\mathbf{F}_{A_p}, \mathbf{F}_p) > \text{Similarity}(\mathbf{F}_i, \mathbf{F}_p)\right) \tag{8}$$

where $|*|$ is the number of items in set $*$, $A$ and $P$ are algorithm and problem sets, $A_p$ denotes the best-performance algorithm for problem $p$, and $[\![A - \{A_p\}]\!]_\beta$ represents the first $\beta|A - \{A_p\}|$ algorithms in the rearrangement of the negative sample in descending order according to the similarity score. For ease of analysis, we replace $\mathbb{I}\left(\text{Similarity}(\mathbf{F}_{A_p}, \mathbf{F}_p) > \text{Similarity}(\mathbf{F}_i, \mathbf{F}_p)\right)$ with $\mathcal{L}(p, i) = \log[1 + e^{-(\text{Similarity}(\mathbf{F}_{A_p}, \mathbf{F}_p) - \text{Similarity}(\mathbf{F}_i, \mathbf{F}_p))}]$, to transform Eq. (8) into a continuous minimization problem as follows:

$$\min \frac{1}{|P|} \sum_{p \in P} \frac{1}{|A - \{A_p\}|} \sum_{i \in [\![A - \{A_p\}]\!]_\beta} \mathcal{L}(p, i) \tag{9}$$

Eq.(9) can be transformed into a distributionally robust optimization (DRO) problem. According to (Zhai et al., 2021), different divergence functions derive different DRO risks. In the following, we choose the commonly-used KL-divergence and CVaR-divergence respectively to deduce a suitable sampling methods. According to the (Zhu et al., 2022), if we use KL-divergence to measure the distance between the worst-case distribution and the uniform distribution, Eq.(9) can be transformed into a distributionally robust optimization problem as:

$$\min_{\lambda \geq 0} \min \frac{1}{|P|} \sum_{p \in P} \left\{ \lambda_p \log E_{i \sim \mathbb{P}}\left[\exp\left(\frac{\mathcal{L}(p, i)}{\lambda_p}\right)\right] + \lambda_p \rho \right\} \tag{10}$$

Let $F(\lambda_p)$ denote the objective function in Eq. (10), then we take the derivative of the variable $\lambda_p$:

$$\frac{\partial F}{\partial \lambda_p} = \rho + \log E_{i \sim \mathbb{P}}\left[\exp\left(\frac{\mathcal{L}(p, i)}{\lambda_p}\right)\right] + \lambda_p \frac{E_{i \sim \mathbb{P}}\left[\exp\left(\frac{\mathcal{L}(p, i)}{\lambda_p}\right) \mathcal{L}(p, i)\right]}{E_{i \sim \mathbb{P}}\left[\exp\left(\frac{\mathcal{L}(p, i)}{\lambda_p}\right)\right](-\lambda_p^2)} \tag{11}$$

Let $\frac{\partial F}{\partial \lambda_p} = 0$ and we obtain the solution $\lambda_p^*$ of Eq. (11) :

$$\lambda_p^* = \frac{E_{i \sim \mathbb{P}}\left[\exp\left(\frac{\mathcal{L}(p, i)}{\lambda_p^*}\right) \mathcal{L}(p, i)\right]}{E_{i \sim \mathbb{P}}\left[\exp\left(\frac{\mathcal{L}(p, i)}{\lambda_p^*}\right)\right] \left\{ \rho + \log E_{i \sim \mathbb{P}}\left[\exp\left(\frac{\mathcal{L}(p, i)}{\lambda_p^*}\right)\right] \right\}} \tag{12}$$

Substitute Eq. (12) into Eq. (10), Eq. (12) can be transformed into:

$$\min \frac{1}{|P|} \sum_{p \in P} \frac{E_{i \sim \mathbb{P}}\left[\exp\left(\frac{\mathcal{L}(p, i)}{\lambda_p^*}\right) \mathcal{L}(p, i)\right]}{E_{i \sim \mathbb{P}}\left[\exp\left(\frac{\mathcal{L}(p, i)}{\lambda_p^*}\right)\right]} \tag{13}$$

Since $E_{i \sim \mathbb{P}}\left[\exp\left(\frac{\mathcal{L}(p, i)}{\lambda_p^*}\right)\right]$ can be taken as a constant, we can obtain the sampling probability for each negative sample by comparing Eq. (9) with Eq. (13):

$$\mathbb{P}(i|p) \propto \frac{\exp\left(\frac{\mathcal{L}(p, i)}{\lambda_p^*}\right)}{E_{i \sim \mathbb{P}}\left[\exp\left(\frac{\mathcal{L}(p, i)}{\lambda_p^*}\right)\right]} \propto \exp\left(\frac{\mathcal{L}(p, i)}{\lambda_p^*}\right) \tag{14}$$

However, we can not provide a quantitative analytical solution for $\lambda_p^*$ here to further determine the analytical solution for $\mathbb{P}(i|p)$ and guide the training process because this is often inaccurate. The

main concern for errors in this context stem from two aspects. Firstly, whether the second derivative of $F(\lambda_p)$ is always positive is unknown, which makes it unfavorable to directly solve for $\lambda_p^*$ from Eq. (12). Secondly, if we use a Taylor expansion to approximate Eq. (10), a hasty choice of the expansion point can result in significant bias. Now, we switch to another CVaR divergence. According to the (Zhai et al., 2021), Eq.(9) can be transformed into another DRO problem as:

$$\min_{\eta_p \geq 0} \min \frac{1}{|P|} \sum_{p \in P} \left\{ \alpha^{-1} E_{i \sim \mathbb{P}} \left[ (\mathcal{L}(p,i) - \eta_p)_+ \right] + \eta_p \right\} \tag{15}$$

According to (Zhai et al., 2021), the optimal $\eta_p$ is the $\alpha$-quantile of $\mathcal{L}(p,i)$:

$$\eta_p^* = \inf_{\eta_p} \left\{ P_{i \sim \mathbb{P}} \left[ \mathcal{L}(p,i) > \eta_p \right] < \alpha \right\} \tag{16}$$

Substitute $P_{i \sim \mathbb{P}} \left[ \mathcal{L}(p,i) > \eta_p^* \right] = \alpha$ into Eq. (15), Eq. (15) can be transformed into:

$$\min \frac{1}{|P|} \sum_{p \in P} \frac{1}{\left| \llbracket A - \{A_p\} \rrbracket_{\mathcal{L}(p,i) > \eta_p^*} \right|} \sum_{i \in \llbracket A - \{A_p\} \rrbracket_{\mathcal{L}(p,i) > \eta_p^*}} \mathcal{L}(p,i) \tag{17}$$

Eq. (17) is consistent with Eq. (9). Hence, Eq. (17) presents a simpler and theoretically reliable sampling method, suggesting the utilization of a uniform distribution for sampling eligible negative instances, i.e., $\mathcal{L}(p,i) > \eta_p^*$. When dealing with a limited number of samples, it is advisable to set $\eta_p^*$ as generously as possible, while in cases with a surplus of negative samples, $\eta_p^*$ can be tightened to enhance sampling precision. This method under uniform distribution is adopted in this paper.

## 4 EXPERIMENTS

**Performance Comparison for Algorithm Selection Tasks**: To verify the performance of AS-LLM, we selected five comparing algorithm selection methods, including Algorithm recommender for white-box problems (AR-WB) (Tian et al., 2020), Implicit multi-fidelity algorithm selection (IMFAS) (Mohan et al., 2022), Algorithm selection for multi-objective optimization problems (AS-MOP) (Tian et al., 2019), Algorithm selection for faster Top-K retrieval (AS-TOPK) (Tolosa & Mallia, 2023), and Task-agnostic representation optimized for algorithm selection (TRIO) (Cohen-Shapira & Rokach, 2022). The training data utilized in this study is sourced from the literature (Tian et al., 2020), denoted as Problemset #$i(i \in 1, 2, \ldots, 20)$. The description of comparing methods and training data can be found in **Appendix A**. Since comparing methods do not require algorithm features, we utilize features extracted from the objective function and employ their learning models to establish a mapping from problem features to algorithms. For AS-LLM, algorithm text serves as another input information, from which LLM extracts features for each algorithm. The code used in this experiment is the original MATLAB code that generates performance data, with redundant data preprocessing stages removed from each code text, retaining only the corresponding algorithm functions. Aside from experiments comparing the code representation capabilities of different LLMs, all other experiments use CodeBERT (Chen et al., 2021) as the LLM. Other detailed parameter settings and experimental process are included in **Appendix A**.

Table 1: Performance of different algorithm selection methods on 10,000 problem instances.

| Algorithm | TRIO | AS-TOPK | AS-MOP | AR-WB | IMFAS | AS-LLM |
|---|---|---|---|---|---|---|
| Problemset #1 | 0.9161 | 0.8814 | 0.8346 | 0.9275 | 0.9271 | **0.9304** |
| Problemset #2 | 0.8021 | 0.7585 | 0.6255 | 0.8219 | 0.8348 | **0.9357** |
| Problemset #3 | 0.8145 | 0.7568 | 0.6953 | 0.8640 | 0.8651 | **0.9554** |
| Problemset #4 | 0.8704 | 0.8391 | 0.7625 | **0.8751** | 0.8720 | 0.8639 |
| Problemset #5 | 0.9054 | 0.8573 | 0.8031 | 0.9168 | 0.9205 | **0.9319** |
| Problemset #6 | 0.8964 | 0.8541 | 0.7923 | 0.9157 | 0.9120 | **0.9244** |
| Problemset #7 | 0.8122 | 0.7680 | 0.6635 | 0.8234 | 0.8362 | **0.8617** |
| Problemset #8 | 0.8112 | 0.8135 | 0.7782 | 0.9004 | 0.8941 | **0.9126** |
| Problemset #9 | 0.8383 | 0.7912 | 0.7163 | 0.8650 | 0.8563 | **0.8788** |
| Problemset #10 | 0.8778 | 0.8235 | 0.7954 | 0.8921 | 0.8916 | **0.9123** |

Tables 1 and 2 demonstrate the average accuracy of decision-making by AS-LLM and each comparing method. In both Table 1 and Table 2, we can observe that AS-LLM consistently outperforms the other algorithm selection methods in terms of accuracy across most of the problem sets. This

Table 2: Performance of different algorithm selection methods on 30,000 problem instances.

| Algorithm | TRIO | AS-TOPK | AS-MOP | AR-WB | IMFAS | AS-LLM |
|---|---|---|---|---|---|---|
| Problemset #11 | 0.9448 | 0.9499 | 0.8423 | 0.9587 | 0.9523 | **0.9662** |
| Problemset #12 | 0.8963 | 0.8897 | 0.6308 | 0.8931 | 0.8993 | **0.9658** |
| Problemset #13 | 0.9135 | 0.8987 | 0.7042 | 0.9135 | 0.9167 | **0.9850** |
| Problemset #14 | 0.9218 | 0.9301 | 0.7670 | 0.9090 | 0.8650 | **0.9280** |
| Problemset #15 | 0.9396 | 0.9412 | 0.7988 | 0.9287 | 0.9269 | **0.9681** |
| Problemset #16 | 0.9361 | 0.9392 | 0.7954 | 0.9369 | 0.9296 | **0.9513** |
| Problemset #17 | 0.9026 | 0.9056 | 0.6602 | 0.8768 | 0.8699 | **0.9113** |
| Problemset #18 | 0.9325 | 0.9297 | 0.7777 | 0.9441 | 0.9106 | **0.9588** |
| Problemset #19 | 0.9096 | 0.9161 | 0.7113 | 0.9114 | **0.9220** | 0.8969 |
| Problemset #20 | 0.9286 | 0.9249 | 0.8031 | 0.9283 | 0.9271 | **0.9578** |

trend indicates the robustness and effectiveness of AS-LLM in handling both smaller (10,000 problem instances) and larger (30,000 problem instances) tasks. We can also observe variations in the performance of other methods. TRIO, AS-TOPK, and IMFAS show competitive results in specific datasets, especially on Problemset #4 with 10,000 problems and Problemset #9 with 30,000problems. By comparing the results from the two tables, we have discovered that AS-LLM gains a significant performance advantage by leveraging additional information sources, i.e., algorithm features, particularly when the training data is limited. For example, when 10,000 samples are insufficient to model the mapping from problem features to algorithm features, AS-LLM, with the assistance of algorithm features, can converge to a superior model more effectively. The results suggest that AS-LLM's ability to leverage algorithm-specific features and model bidirectional relationships in algorithm selection is particularly advantageous. It exhibits robustness across various datasets, making it a reliable choice for algorithm selection tasks, especially especially it is difficult to directly model the mapping from problem features to algorithms under low-quality training data.

Table 3: Ablation Experiment on 30,000 problem instances (**Appendix B** provides more results).

| | AS-LLM-Si | AS-LLM-AF | AS-LLM-MLP | AS-LLM-Pool | AS-LLM |
|---|---|---|---|---|---|
| Problemset #11 | 0.948 | 0.802 | 0.949 | 0.952 | **0.953** |
| Problemset #12 | 0.968 | 0.402 | 0.974 | 0.979 | **0.982** |
| Problemset #14 | 0.904 | 0.754 | 0.919 | 0.932 | **0.932** |
| Problemset #19 | 0.922 | 0.657 | 0.935 | 0.939 | **0.943** |

**Ablation Experiments**: Ablation experiments are conducted to assess the impact of four modules in AS-LLM: cosine similarity calculations, track for algorithm representation, average pooling layer, and the MLP layer for output. Variant AS-LLM-Si uses both algorithm and problem features but doesn't calculate the matching degree; it directly utilizes an MLP instead. Variant AS-LLM-AF removes entire algorithm track, relying solely on problem features and use feature track directly. Variant AS-LLM-Pool replaces average pooling with maximum pooling. Variant AS-LLM-MLP computes matching degrees but doesn't consider algorithm features in the final MLP layer. Experiments on four variants and the original AS-LLM choose problem sets in different scale of instances and candidate algorithms, to assess module impacts. From Table 3 as well as Table 5 in Appendix B, we conclude that: Compared with AS-LLM, AS-LLM-AF differed significantly, emphasizing the importance of algorithm features and algorithm track. AS-LLM-Si showed substantial performance differences, highlighting the effectiveness of problem-algorithm feature matching. AS-LLM-MLP and AS-LLM-Pool have the least performance loss, but they also demonstrate the positive impact of the MLP layer and average pooling layer on the results. These two modules respectively contribute to better gradient backpropagation and faster convergence, thus positively influencing the model's accuracy. Some more detailed analyses are provided in **Appendix B**.

**Impact of Negative Sampling**: To explore the relationship between the probability of negative sampling and model performance as well as training time, a series of experiments were conducted on Problemset #1 and Problemset #11, which contain the most candidate algorithms. The probability of negative sampling was incrementally increased from $10\%$ to $90\%$, and sampling was performed under a uniform distribution according to Section 3.4. The experiments recorded the best performance achieved within 300 epochs and the average runtime for each epoch. The experimental results are presented in Figure 3 (a) and Figure 1 in **Appendix C**. From the graph, it is evident that as the probability of negative sampling increases, the algorithm's training time also increases sequentially. However, the changes in performance were not consistently increasing with the rise in sampling rates. In both datasets, there were performance degradation phenomenon under an increasing sam-

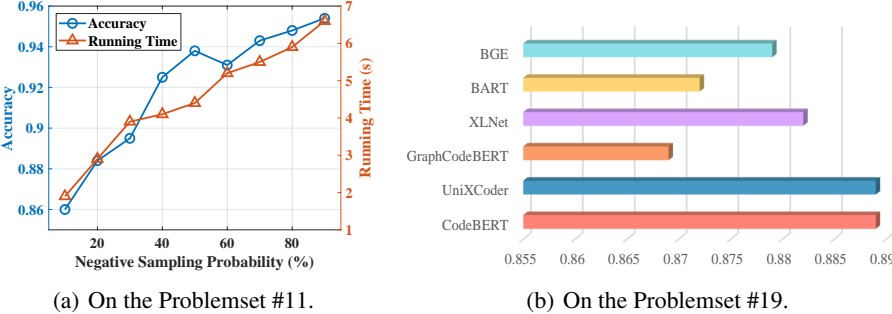

(a) On the Problemset #11.                    (b) On the Problemset #19.

Figure 3: (a) The relationship between the probability of negative sampling and model performance as well as time efficiency (**Appendix C** provides more results). (b) The performance of different LLMs (**Appendix D** provides more results).

pling rate. This suggests that : (1) The choice of negative sampling probability should strike a balance between accuracy and time efficiency. (2) Careful adjustment of the negative sampling probability is essential to avoid performance degradation.

**Evaluating the Code Comprehension Capabilities of LLMs:** LLM assists AS-LLM in extracting algorithm representations from code text. Conversely, the algorithm selection task based on AS-LLM can serve as a benchmark task for evaluating LLM's code comprehension abilities. To validate this assertion, this subsection conducts experiments where different pre-trained LLMs are deployed within AS-LLM's code feature extraction module to compare their performance differences. Two categories of large models participate in this experiment: one comprises general-purpose LLMs trained on massive textual data including XLNet (Yang et al., 2019b), BART (Lewis et al., 2020), and BGE (Xiao et al., 2023), while the other consists of LLMs trained specifically for code-related tasks such as CodeBERT (Chen et al., 2021), GraphCodeBERT (Guo et al., 2020), and UniXCoder (Guo et al., 2022). We tested the six LLMs on problem sets with different numbers of problems and candidate algorithms, as depicted in Figure 3 (b) and Figures 2 and 3 in **Appendix D**. Overall, LLMs pre-trained on code text, such as UniXCoder and CodeBERT, exhibit greater stability, with UniXCoder achieving the best or highly competitive performance on almost all datasets. Although large models pre-trained on common text also possess code representation capabilities and occasionally perform exceptionally well, their stability falls short of LLMs pre-trained on code text. For instance, BART even outperformed all code-pre-trained LLMs on Problemset #12. However, as the number of algorithms increases, LLMs pre-trained on code text exhibit greater advantages. The recently introduced BGE model also demonstrates relatively stable performance in code representation. In summary, this experiment indicates the potential of the AS-LLM-based algorithm selection framework as a benchmark for evaluating LLMs' code representation capabilities.

## 5   CONCLUSION

This paper has explored the crucial yet underrepresented dimension of algorithm features in the domain of algorithm selection. A novel framework, AS-LLM, is proposed that not only leverages information about problem features and algorithm performance but also captures algorithm representations. This approach marks a departure from traditional algorithm selection methods that predominantly focus on problem features. Our investigation has yielded several significant contributions to the field of algorithm selection. Firstly, we have demonstrated the feasibility of extracting algorithm features using pre-trained LLMs. This method offers generality across a wide range of algorithms, potentially revolutionizing the way we approach algorithm selection. Secondly, the AS-LLM framework has showcased practical benefits, including a more aligned approach to the bidirectional nature of algorithm selection tasks, superior performance with fewer required training samples, and versatility to adapt to different problem types. Lastly, the AS-LLM framework opens up new avenues for evaluating the algorithm comprehension capabilities of LLMs. By embedding various LLMs into AS-LLM, we have shed light on performance differences among them in the context of algorithm representation.

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
