# APPENDIX: EXPERIMENTAL DETAILS OF AS-LLM

The experiments focus on the meta-heuristic algorithm selection for continuous optimization problems to validate the effectiveness of AS-LLM, which are divided into three parts. First, we conduct comparative experiments on candidate algorithm sets of varying sizes and different scales of training data, to compare the performance of AS-LLM against the state-of-the-art algorithm selection methods. Subsequently, we perform ablation experiments on the model by replacing or eliminating parts of the model to validate the role of each module. Following this, there is an analysis experiment that primarily examines the impact of negative sampling on the accuracy and time efficiency of AS-LLM. Finally, we embed different LLMs into AS-LLM to showcase and compare the capabilities of these LLMs in code comprehension and representation.

## A. PERFORMANCE COMPARISON FOR ALGORITHM SELECTION TASKS

**Comparing Algorithms**: To verify the performance of AS-LLM, we selected five comparing algorithm selection methods proposed in recent years as follows:

- Algorithm recommender for white-box problems (AR-WB) (Tian et al., 2020). AR-WB is a algorithm selection system for continuous optimization problems. It represents problem structures using tree-based features and utilizes a deep recurrent neural network to recommend the most suitable metaheuristic algorithm.

- Implicit multi-fidelity algorithm selection (IMFAS) (Mohan et al., 2022). IMFAS is a algorithm selection method that directly leverages multi-fidelity landmarking information from candidate algorithms, learning dataset topology and algorithm biases via LSTMs and MLP in a few-shot setting.

- Algorithm selection for multi-objective optimization problems (AS-MOP) (Tian et al., 2019). This paper introduces an automated algorithm selection method for MOPs, which uses a SVM predictor to choose the most suitable evolutionary algorithm based on explicit and implicit features of an MOP.

- Algorithm selection for faster Top-K retrieval (AS-TOPK) (Tolosa & Mallia, 2023). This study introduces a machina learning (decision tree / random forest) based algorithm selection method to predict the time-efficiency of different top-k query processing algorithms.

- Task-agnostic representation optimized for algorithm selection (TRIO) (Cohen-Shapira & Rokach, 2022). TRIO is a meta-learning approach that utilizes learned graphical representations of datasets to recommend the best-performing machine learning algorithms with XGBoost algorithm.

As the aforementioned models were not explicitly designed for algorithm selection in continuous optimization problems, certain algorithms may not be suitable for feature extraction in this context. In such cases, we utilize features extracted from the objective function and employ separate learning models from the comparing methods to establish a mapping from problem features to algorithms.

**Dataset Description**: The training data utilized in this study is sourced from the literature (Tian et al., 2020). The training data collection strategy employed in the cited work involves creating complex functions using a tree-like structure. The initial nodes of the tree consist of mean computation and $x$, after which a random leaf node is selected and extended with a new operator, along with the random selection of new operands based on the operator. To construct more intricate functions, various operations are introduced during the sampling process to modify the constructed tree, including the addition of noise, removal of redundant operations, and substitution of complex functions. After collecting the objective functions for the given problems, candidate algorithm sets of various sizes are deployed on each problem in the problem set, and performance for each candidate

Table 1: Proportion of each best-performance algorithm within 10,000 problem instances.

| Datasets | Proportion of each best-performance algorithm | | | | | | | | | |
|---|---|---|---|---|---|---|---|---|---|---|
| Problemset #1 | Alg1 | Alg2 | Alg3 | Alg4 | Alg5 | Alg6 | Alg7 | Alg8 | Alg9 | Alg10 |
| | 0.26% | 6.62% | 30.44% | 7.59% | 2.59% | 0.08% | 41.30% | 9.38% | 1.67% | 0.07% |
| Problemset #2 | Alg1 | Alg2 | | | | | | | | |
| | 37.19% | 62.81% | | | | | | | | |
| Problemset #3 | Alg1 | Alg2 | | | | | | | | |
| | 70.07% | 29.93% | | | | | | | | |
| Problemset #4 | Alg1 | Alg2 | Alg3 | Alg4 | Alg5 | Alg6 | Alg7 | Alg8 | | |
| | 1.19% | 10.46% | 34.40% | 15.55% | 2.64% | 13.60% | 21.89% | 0.27% | | |
| Problemset #5 | Alg1 | Alg2 | Alg3 | Alg4 | Alg5 | Alg6 | Alg7 | Alg8 | | |
| | 2.65% | 32.48% | 7.63% | 2.64% | 43.48% | 9.58% | 1.47% | 0.07% | | |
| Problemset #6 | Alg1 | Alg2 | Alg3 | Alg4 | Alg5 | Alg6 | Alg7 | Alg8 | | |
| | 6.59% | 30.86% | 7.55% | 2.56% | 0.19% | 41.34% | 9.44% | 1.47% | | |
| Problemset #7 | Alg1 | Alg2 | Alg3 | Alg4 | Alg5 | | | | | |
| | 25.46% | 32.30% | 17.20% | 17.50% | 7.54% | | | | | |
| Problemset #8 | Alg1 | Alg2 | Alg3 | Alg4 | Alg5 | | | | | |
| | 21.42% | 11.91% | 0.84% | 55.48% | 10.25% | | | | | |
| Problemset #9 | Alg1 | Alg2 | Alg3 | Alg4 | Alg5 | | | | | |
| | 42.46% | 9.35% | 34.78% | 12.30% | 1.11% | | | | | |
| Problemset #10 | Alg1 | Alg2 | Alg3 | Alg4 | Alg5 | | | | | |
| | 61.54% | 9.08% | 21.20% | 7.81% | 0.37% | | | | | |

Table 2: Proportion of each best-performance algorithm within 30,000 problem instances.

| Datasets | Proportion of each best-performance algorithm | | | | | | | | | |
|---|---|---|---|---|---|---|---|---|---|---|
| Problemset #11 | Alg1 | Alg2 | Alg3 | Alg4 | Alg5 | Alg6 | Alg7 | Alg8 | Alg9 | Alg10 |
| | 0.20% | 6.50% | 30.50% | 7.81% | 2.41% | 0.10% | 41.63% | 9.18% | 1.57% | 0.08% |
| Problemset #12 | Alg1 | Alg2 | | | | | | | | |
| | 37.02% | 62.98% | | | | | | | | |
| Problemset #13 | Alg1 | Alg2 | | | | | | | | |
| | 70.44% | 29.56% | | | | | | | | |
| Problemset #14 | Alg1 | Alg2 | Alg3 | Alg4 | Alg5 | Alg6 | Alg7 | Alg8 | | |
| | 1.20% | 10.64% | 34.81% | 15.63% | 2.92% | 13.41% | 21.21% | 0.18% | | |
| Problemset #15 | Alg1 | Alg2 | Alg3 | Alg4 | Alg5 | Alg6 | Alg7 | Alg8 | | |
| | 2.63% | 32.24% | 7.85% | 2.67% | 43.55% | 9.44% | 1.56% | 0.06% | | |
| Problemset #16 | Alg1 | Alg2 | Alg3 | Alg4 | Alg5 | Alg6 | Alg7 | Alg8 | | |
| | 6.89% | 30.64% | 7.61% | 2.51% | 0.19% | 41.17% | 9.44% | 1.55% | | |
| Problemset #17 | Alg1 | Alg2 | Alg3 | Alg4 | Alg5 | | | | | |
| | 25.26% | 32.57% | 17.42% | 17.07% | 7.68% | | | | | |
| Problemset #18 | Alg1 | Alg2 | Alg3 | Alg4 | Alg5 | | | | | |
| | 21.53% | 11.47% | 0.90% | 55.51% | 10.59% | | | | | |
| Problemset #19 | Alg1 | Alg2 | Alg3 | Alg4 | Alg5 | | | | | |
| | 43.12% | 9.74% | 34.05% | 12.07% | 1.02% | | | | | |
| Problemset #20 | Alg1 | Alg2 | Alg3 | Alg4 | Alg5 | | | | | |
| | 60.66% | 9.33% | 21.41% | 8.25% | 0.35% | | | | | |

algorithm on each problem are obtained. We selected candidate algorithm sets of different sizes, denoted as Problemset $\#i (i \in 1, 2, \ldots, 20)$, each containing different candidate algorithms. Tables 1 and 2 provides the number of algorithms included in each set and the proportion of each algorithms that achieved the best performance.

**Experimental Settings**: In addition to obtaining the aforementioned problem features, algorithm text serves as another input information for the AS-LLM algorithm, from which LLM extracts features for each algorithm. The code used in this experiment is the original MATLAB code that generates performance data, with redundant data preprocessing stages removed from each code text, retaining only the corresponding algorithm functions. Aside from experiments comparing the code representation capabilities of different LLMs, all other experiments use CodeBert (Chen et al., 2021) as the LLM. Since most LLMs have high-dimensional representations for code text, for example, CodeBert extracts features with dimensions ranging from hundreds of thousands to millions, a two-stage pooling layer is employed to reduce the extremely high-dimensional algorithm features in order to control their impact on the results. The final problem and algorithm representations are set to the same 10 dimensions. The loss function used is cross-entropy loss, which assesses the model's predictions of the positivity or negativity of each example. The training batch size for all deep models is set to 128. In the TRIO algorithm, the maximum tree depth for XGBoost is set to 100, and the number of weak classifiers is set to 1000. The SVM and decision tree classifiers used in the AS-MOP and AS-TOPK algorithms respectively are configured with default settings from sklearn.

Table 3: Performance of different algorithm selection methods on 10000 problem instances.

| Algorithm | TRIO | AS-TOPK | AS-MOP | AR-WB | IMFAS | AS-LLM |
|---|---|---|---|---|---|---|
| Problemset #1 | 0.9161 | 0.8814 | 0.8346 | 0.9275 | 0.9271 | **0.9304** |
| Problemset #2 | 0.8021 | 0.7585 | 0.6255 | 0.8219 | 0.8348 | **0.9357** |
| Problemset #3 | 0.8145 | 0.7568 | 0.6953 | 0.864 | 0.8651 | **0.9554** |
| Problemset #4 | 0.8704 | 0.8391 | 0.7625 | **0.8751** | 0.8720 | 0.8639 |
| Problemset #5 | 0.9054 | 0.8573 | 0.8031 | 0.9168 | 0.9205 | **0.9319** |
| Problemset #6 | 0.8964 | 0.8541 | 0.7923 | 0.9157 | 0.9120 | **0.9244** |
| Problemset #7 | 0.8122 | 0.7680 | 0.6635 | 0.8234 | 0.8362 | **0.8617** |
| Problemset #8 | 0.8112 | 0.8135 | 0.7782 | 0.9004 | 0.8941 | **0.9126** |
| Problemset #9 | 0.8383 | 0.7912 | 0.7163 | 0.8650 | 0.8563 | **0.8788** |
| Problemset #10 | 0.8778 | 0.8235 | 0.7954 | 0.8921 | 0.8916 | **0.9123** |

Table 4: Performance of different algorithm selection methods on 30000 problem instances.

| Algorithm | TRIO | AS-TOPK | AS-MOP | AR-WB | IMFAS | AS-LLM |
|---|---|---|---|---|---|---|
| Problemset #11 | 0.9448 | 0.9499 | 0.8423 | 0.9587 | 0.9523 | **0.9662** |
| Problemset #12 | 0.8963 | 0.8897 | 0.6308 | 0.8931 | 0.8993 | **0.9658** |
| Problemset #13 | 0.9135 | 0.8987 | 0.7042 | 0.9135 | 0.9167 | **0.9850** |
| Problemset #14 | 0.9218 | 0.9301 | 0.7670 | 0.9090 | 0.8650 | **0.9280** |
| Problemset #15 | 0.9396 | 0.9412 | 0.7988 | 0.9287 | 0.9269 | **0.9681** |
| Problemset #16 | 0.9361 | 0.9392 | 0.7954 | 0.9369 | 0.9296 | **0.9513** |
| Problemset #17 | 0.9026 | 0.9056 | 0.6602 | 0.8768 | 0.8699 | **0.9113** |
| Problemset #18 | 0.9325 | 0.9297 | 0.7777 | 0.9441 | 0.9106 | **0.9588** |
| Problemset #19 | 0.9096 | 0.9161 | 0.7113 | 0.9114 | **0.9220** | 0.8969 |
| Problemset #20 | 0.9286 | 0.9249 | 0.8031 | 0.9283 | 0.9271 | **0.9578** |

We conducted multiple rounds of testing on datasets of varying sizes, with $80\%$ of the samples in each dataset allocated as training data and the remaining $20\%$ as test data.

**Performance Evaluation**: Tables 3 and 4 demonstrate the average accuracy of decision-making by AS-LLM and each comparing method. In both Table 3 and Table 4, we can observe that AS-LLM consistently outperforms the other algorithm selection methods in terms of accuracy across most of the problem sets. This trend indicates the robustness and effectiveness of AS-LLM in handling both smaller (10,000 problem instances) and larger (30,000 problem instances) tasks. We can also observe variations in the performance of other methods. TRIO, AS-TOPK, and IMFAS show competitive results in specific datasets, especially on Problemset #4 with 10,000 problems and Problemset #9 with 30,000problems. By comparing the results from the two tables, we have discovered that AS-LLM gains a significant performance advantage by leveraging additional information sources, i.e., algorithm features, particularly when the training data is limited. For example, when 10,000 samples are insufficient to model the mapping from problem features to algorithm features, AS-LLM, with the assistance of algorithm features, can converge to a superior model more effectively. The results suggest that AS-LLM's ability to leverage algorithm-specific features and model bidirectional relationships in algorithm selection is particularly advantageous. It exhibits robustness across various datasets, making it a reliable choice for algorithm selection tasks, especially especially it is difficult to directly model the mapping from problem features to algorithms under low-quality training data.

## B. ABLATION EXPERIMENTS

In this subsection, we conduct ablation experiments to assess the impact of the four modules: (1) the significance of cosine similarity calculations, (2) the effectiveness of algorithm track for algorithm representation, (3) the efficacy of average pooling layer in algorithm track, and (4) the function of the MLP layer used for output generation. Specifically, AS-LLM represents the original model, while its four variant models are defined as follows: (1) AS-LLM-Si utilizes both algorithm and problem features but does not calculate the degree of matching degree between algorithm and problem representations; instead, it passes the representations of algorithms and problems through an MLP layer. (2) AS-LLM-AF removes algorithm track from the AS-LLM model, retaining only problem features, and directly learns the mapping from problem representations. (3) AS-LLM-Pool replaces the average pooling in AS-LLM with a maximum pooling layer. (4) AS-LLM-MLP employs both algorithm and problem features and computes the matching degree between algorithm and problem representations, but the final MLP layer does not consider algorithm features.

Table 5: Ablation Experiment on 10,000 problem instances.

|  | AS-LLM-Si | AS-LLM-AF | AS-LLM-MLP | AS-LLM-Pool | AS-LLM |
|---|---|---|---|---|---|
| Problemset #1 | 0.918 | 0.797 | 0.929 | 0.929 | **0.931** |
| Problemset #2 | 0.944 | 0.341 | 0.945 | 0.945 | **0.945** |
| Problemset #4 | 0.836 | 0.749 | 0.866 | 0.869 | **0.872** |
| Problemset #9 | 0.857 | 0.656 | 0.888 | 0.887 | **0.892** |

Table 6: Ablation Experiment on 30,000 problem instances.

|  | AS-LLM-Si | AS-LLM-AF | AS-LLM-MLP | AS-LLM-Pool | AS-LLM |
|---|---|---|---|---|---|
| Problemset #11 | 0.948 | 0.802 | 0.949 | 0.952 | **0.953** |
| Problemset #12 | 0.968 | 0.402 | 0.974 | 0.979 | **0.982** |
| Problemset #14 | 0.904 | 0.754 | 0.919 | 0.932 | **0.932** |
| Problemset #19 | 0.922 | 0.657 | 0.935 | 0.939 | **0.943** |

We choose 8 problem sets with different numbers of problems and candidate algorithms, to conduct experiments on four variants of AS-LLM and the original AS-LLM and comprehensively and fairly assess the impact of each module. The results of the ablation experiments are presented in Tables 5 and 6. (1) The variant with the most significant difference from AS-LLM is AS-LLM-AF. It exhibited the poorest performance in all experiments when algorithm features were omitted. This underscores the crucial influence of algorithm features and algorithm track on algorithm selection problems. (2) AS-LLM-Si displayed a substantial performance difference compared to AS-LLM, indicating that the matching degree between problem features and algorithm features in our model is effective and has a significant impact on the final results. (3) AS-LLM-MLP showed performance differences compared to AS-LLM, but the disparities were not as pronounced as in the first two variants. This module is not a core component of the model, as it is placed after similarity computation and mainly facilitates gradient backpropagation, thus positively influencing the model's accuracy but exerting a limited impact on it. (4) The variant with the smallest difference from AS-LLM is AS-LLM-Pool. Despite selecting the performance-enhancing average pooling, different pooling methods had less noticeable effects on performance in the algorithm selection scenario. The main reason is the preprocessing of high-dimensional algorithm features, which diminishes the differences between various pooling methods.

In summary, the core contributions highlighted in this study, including algorithm representation and modeling the matching degree between problems and algorithms, had a significant impact on the model's results. AS-LLM accurately collects training information through problem track and algorithm track, effectively models the matching mechanism between problems and algorithms through similarity calculation, and thus demonstrates the state-of-the-art performance.

## C. IMPACT OF NEGATIVE SAMPLING

To explore the relationship between the probability of negative sampling and model performance as well as training time, a series of experiments were conducted on Problemset #1 and Problemset #11, which contain the most candidate algorithms. The probability of negative sampling was incrementally increased from 10% to 90%, and sampling was performed under a uniform distribution according to Section 3.4 in the main text. The experiments recorded the best performance achieved within 300 epochs and the average runtime for each epoch. The experimental results are presented in Figure 1.

From the graph, it is evident that as the probability of negative sampling increases, the algorithm's training time also increases sequentially. This is attributed to the higher probability resulting in the generation of more samples, thereby increasing training time. However, the changes in performance were not consistently increasing with the rise in sampling rates. In both datasets, there were performance degradation phenomenon under an increasing sampling rate. This suggests that the impact of sampling rates on accuracy should be considered in conjunction with various factors, including the specific scenario, the scale of training data, the quality of the samples, and so on. From this experiment, two conclusions can be drawn: (1) The choice of negative sampling probability should strike a balance between accuracy and time efficiency. (2) Careful adjustment of the negative sampling probability is essential to avoid performance degradation.

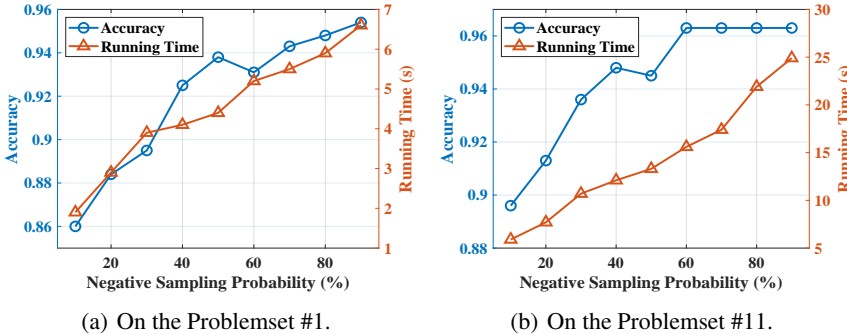

(a) On the Problemset #1.  (b) On the Problemset #11.

Figure 1: The relationship between the probability of negative sampling and model performance as well as time efficiency.

## D. EVALUATING THE CODE COMPREHENSION CAPABILITIES OF LLMS

LLM assists AS-LLM in extracting algorithm representations from code text. Conversely, the algorithm selection task based on AS-LLM can serve as a benchmark task for evaluating LLM's code comprehension abilities. To validate this assertion, this subsection conducts experiments where different pre-trained LLMs are deployed within AS-LLM's code feature extraction module to compare their performance differences. Two categories of large models participate in this experiment: one comprises general-purpose LLMs trained on massive textual data (including code text and other text), while the other consists of LLMs trained specifically for code-related tasks. Their respective descriptions are as follows:

(1) General-purpose LLMs:

- XLNet (Yang et al., 2019): XLNet is a self-regressive pre-trained language model that employs the permutation language modeling approach, which allows for better handling of long texts and multi-document tasks. Its primary functionalities include text generation, text classification, and question-answering systems, among others.

- BART (Lewis et al., 2020): BART is a sequence-to-sequence generative model that utilizes both autoregressive and autoencoding methods, making it suitable for various tasks such as text summarization, translation, and dialogue generation.

- BGE (Xiao et al., 2023): BGE is a recently released large-scale pre-trained model based on natural language processing. It has demonstrated superior performance in various tasks, including text generation, text classification, and question-answering systems, in both Chinese and English.

(2) Code-specific LLMs trained for code-related tasks:

- CodeBERT (Chen et al., 2021): CodeBERT is a pre-trained model specifically designed for source code. It can be applied to tasks like code completion, code recommendation, and code searching.

- GraphCodeBERT (Guo et al., 2020): GraphCodeBERT is a model that combines graph neural networks with self-regressive pre-training. It can be used for tasks like code completion, code recommendation, and code searching.

- UniXCoder (Guo et al., 2022): UniXcoder is a unified cross-modal pre-trained model that leverages multimodal data (i.e. code comment and abstract syntax tree) to pre-train code representation.

It is reasonable to evaluate LLMs using the algorithm selection framework based on AS-LLM because the algorithm selection process can take into account code text from various programming language and across multiple algorithms. We tested the six LLMs on datasets with different candidate algorithm scales, as depicted in Figures 2 and 3. Overall, LLMs pre-trained on code text,

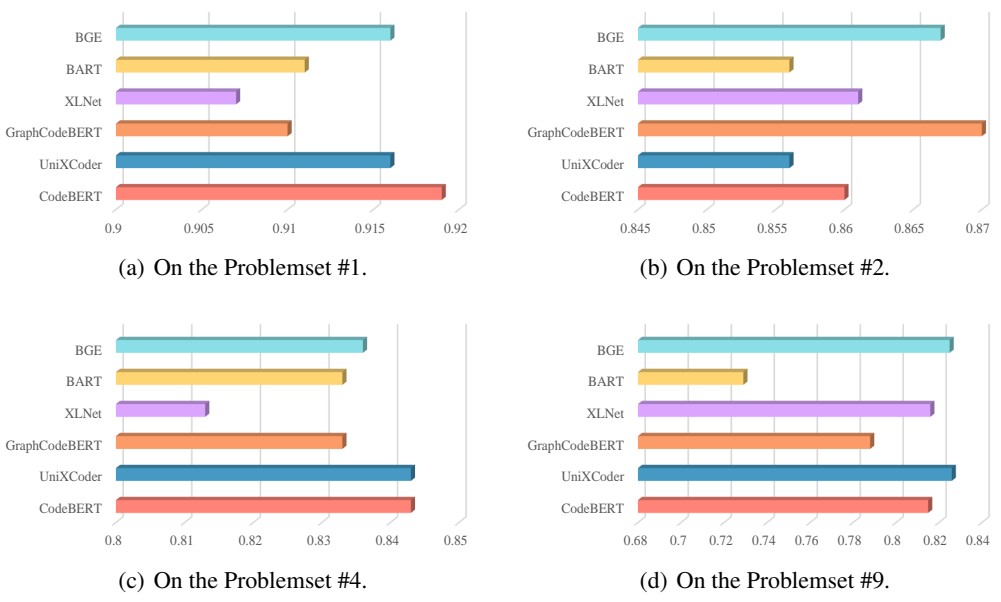

Figure 2: The performance of different LLMs on 10,000 problem instances.

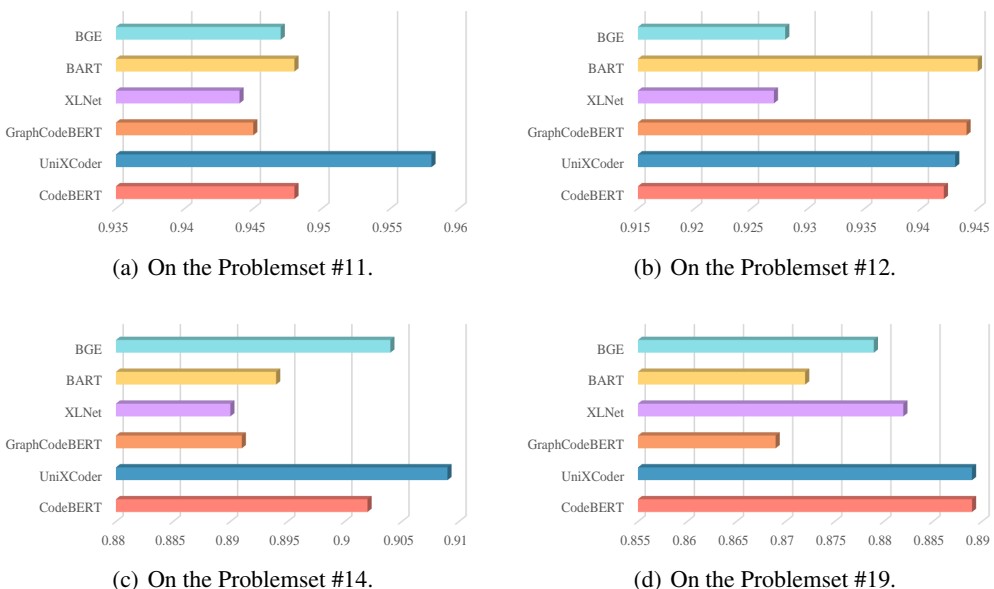

Figure 3: The performance of different LLMs on 30,000 problem instances.

such as UniXCoder and CodeBERT, exhibit greater stability, with UniXCoder achieving the best or highly competitive performance on almost all datasets. Although large models pre-trained on common text also possess code representation capabilities and occasionally perform exceptionally well, their stability falls short of LLMs pre-trained on code text. For instance, BART even outperformed all code-pre-trained LLMs on Problemset #12. However, as the number of algorithms increases, LLMs pre-trained on code text exhibit greater advantages. The recently introduced BGE model also demonstrates relatively stable performance in code representation. In summary, this experiment indicates the potential of the AS-LLM-based algorithm selection framework as a benchmark for evaluating LLMs' code representation capabilities.