# OpenReview forum: "AS-LLM: When Algorithm Selection Meets Large Language Model"
_ICLR.cc/2024/Conference — ICLR 2024 Conference Withdrawn Submission_

### Official Review · Reviewer_KfZ1 · 2023-10-19

**Soundness:** 2 fair
**Presentation:** 2 fair
**Contribution:** 2 fair
**Rating:** 3
**Confidence:** 4

**Summary:**

The authors suggest a framework incorporating algorithm features extracted from algorithm source code into algorithm selection. In particular, they leverage LLMs and their code comprehension capabilities to generate such features and show that their framework performs well to some competitors on a set of continuous optimization problems found in the literature.

**Strengths:**

* The work is original in the sense that the authors connect LLMs and algorithm selection. While this has been done for other AutoML subproblems, it has not been done for algorithm selection to the best of my knowledge. I believe that, in general, such a connection offers very interesting avenues for further research.
* The paper is very clearly written overall and in general easy to understand.
* If the algorithm features found by the authors are meaningful and at the same time easy to compute, there is a good chance that they can increase the applicability of algorithm selection in practice.

**Weaknesses:**

The paper suffers from a variety of problems from my perspective which I outline below:

1) From the originality/novelty perspective, in contrast to the claim made by the authors (p. 2), their paper is not the first one to suggest the idea of algorithm features as part of algorithm selection. Although literature on the topic is rather sparse, there are a bunch of works, which have done so. See [A, B], for example. [C] even provides an overview of related work regarding algorithm features in algorithm selection in Section 3.5. Notably, [B] even computes algorithm features based on source code analysis and thus is extremely similar to the work of the authors, but never mentioned. The main methodological difference between [B] and this paper is that the authors here use an LLM to extract code features, which from a contribution perspective is rather weak on its own for a conference such as ICLR, in my opinion.
2) From a quality perspective, the evaluation of the work suffers from a set of problems making any conclusions hard to draw:
    1) The benchmark used by the authors is very uncommon in algorithm selection. The standard benchmark in algorithm selection is ASLib [D] and for the community it is very hard to assess how much improvement the algorithm features extracted by the LLM actually help due to that as they cannot set it into the context of their well known benchmark.
    2) The competitors chosen by the authors are not only rather unknown in the algorithm selection community, but at their core some of them have significantly other goals than the approach of the author making it an extremely unfair comparison from my perspective. For example, AS-MOP is an algorithm selector for multi-objective problems. However, I do not see how the approach of the authors can be used for multi-objective problems nor do I think that the benchmark has multi-objective components (sorry if I missed that). Similarly, IMFAS was built for selecting machine learning algorithms that allow to estimate their performance based on fidelities. I am not even sure how the authors adapted the approach to their problem as this is certainly not straightforward. Unfortunately, this is also not discussed in the appendix. Besides, IMFAS is the predecessor of MASIF [E] such that rather MASIF should be used (although I doubt that any of the two is a reasonable competitor for the benchmark the authors chose). Overall, the competitors are mostly unsuitable to compare against and much more importantly it is unclear how and if they were adapted to the problem of the authors.
    3) None of the evaluations carry any uncertainty information such as the variance of the results. Considering that many of the approaches behave very similarly in terms of performance (Tables 2 or 3), the "improvement" shown by the authors could essentially just be noise. That being said, the experiments lack in details regarding the number of repetitions that were performed, the hardware that was used and even the concrete evaluation function that was used. The authors state Table 1 and 2 "demonstrate the average accuracy of decision-making by AS-LLM..." (p. 7), but never formally define what this function concretely looks like. Since the authors also do not release any source code with this submission, there is no way to verify any of this myself.
Overall, the experimental results do not look very convincing to me and do not support any of the claims made by the authors.
3) The authors leverage an LSTM to generate instance features as they state on page 5. I wonder why no transformer architecture was used since they are generally assumed to be more capable than LSTMS.
4) The notation used is a bit inconsistent across the paper. At the beginning, an algorithm is denoted by $A_j$ and an instance by $I_i$. In Section 3.4, however, algorithms are sometimes just denoted by the index (which is an $i$ suddenly) and instances by $p$.
5) There is already quite some work in the intersection between LLMs and AutoML (including algorithm selection as a subtopic), which is not discussed at all. The authors should at least acknowledge that these works exists and that they are related to this paper. See [F] for an overview on the intersection of the topics.
6) The authors make various statements throughout the paper that, as an expert in algorithm selection, I cannot support or see at least critical:
    1) p. 3: "[...] algorithm selection is fundamentally straightforward." -> If the was the case, there would not be research for decades on the topic.
    2) p. 2: "Firstly,  disregarding  algorithm  features as  an  essential  information  source  inevitably  results  in  a  loss  of  model  accuracy" -> This does not have to be the case. One can construct examples, where the algorithm features will not serve any benefits. This statement should be phrased more carefully.
    3) p. 2: "Requiring substantial and hard-to-acquire training data, such as performance data across diverse problems, undermines the original intent of algorithm selection" -> I disagree here. AS tries to find the most suitable algorithm for a given instance. If we need to mine data to be able to do that once upfront, I do not see how that undermines the original intent of AS. Moreover, the same problem still applies to the approach suggested by the authors. In fact they do not provide any analysis that with their approach, less training data is required, but rather that one gets a better AS performance with their approach.
    4) p. 2: "To the best of our knowledge, this paper pioneers the integration of algorithm features into the algorithm selection process [...]" -> As mentioned above, this paper is not a pioneer in that regard.
7) Minor problems:
    1) Abstract: "AutoML" is introduced without explaining the abbreviation.
    2) p. 3: "a algorithm set" -> "an algorithm set"
    3) p. 3: "significantly impact the performance" -> "significantly impacts the performance"
    4) The background section quickly discussed different models for algorithm selection, but misses quite some. In particular, ranking ones, hybrid regression and ranking ones and also survival analysis ones. [C] also gives a good overview on this in Section 2.3.
    5) p. 4: "provide accurate match" -> "provide an accurate match"


[A] Tornede, Alexander, et al. "Extreme algorithm selection with dyadic feature representation." International Conference on Discovery Science. Cham: Springer International Publishing, 2020.

[B] Pulatov, Damir, et al. "Opening the Black Box: Automated Software Analysis for Algorithm Selection." International Conference on Automated Machine Learning. PMLR, 2022.

[C] Tornede, Alexander. "Advanced Algorithm Selection with Machine Learning: Handling Large Algorithm Sets, Learning From Censored Data, and Simplifying Meta Level Decisions.", 2023

[D] Bischl, Bernd, et al. "Aslib: A benchmark library for algorithm selection." Artificial Intelligence 237 (2016): 41-58.

[E] Ruhkopf, Tim, et al. "MASIF: Meta-learned Algorithm Selection using Implicit Fidelity Information." Transactions on Machine Learning Research (2022).

[F] Tornede, Alexander, et al. "AutoML in the Age of Large Language Models: Current Challenges, Future Opportunities and Risks." arXiv preprint arXiv:2306.08107 (2023).

**Questions:**

* On p. 5 the authors write "However, when incorporating algorithm features into AS-LLM, a reconstruction of training samples becomes necessary, with a critical focus on sampling of negative instances" -> Why is this only the case in the algorithm feature case? You can also train a multi-class classification system based on problem instances only and whatever considerations one takes when algorithm features are available, should also be taken when they are not available, imo. Please elaborate on this more.
* I also wonder why no simple multi-class classification model or even more advanced a regression model was used by the authors as a last step of the pipeline. The latter would certainly dismiss the need for carefully choosing negative samples.
* I would find it extremely interesting, if the authors analyzed the features extracted by the LLM for the usage of AS a bit more. What are feature importance values of these algorithm features? What role do they concretely play in the decision on the final algorithm? How do these features look like? Can you cluster similar algorithms in that space based on the extracted features?

In order to make me change my rating, the authors have to fix all of the weaknesses I mentioned above. Most importantly, (1) I would like to see a truly meaningful evaluation of their approach on a known benchmark to fair competitors. (2) Moreover, I would like to see the work put into context better including existing work on algorithm features in AS and the intersection between LLMs and AutoML.

To be very honest, these requested changes will drastically change the paper requiring a completely new review, from my perspective.

---

### Official Review · Reviewer_R4Vs · 2023-10-27

**Soundness:** 2 fair
**Presentation:** 2 fair
**Contribution:** 2 fair
**Rating:** 3
**Confidence:** 4

**Summary:**

In the paper "AS-LLM: When Algorithm Selection Meets Large Language Model" the authors propose to augment the set of problem features by a set of algorithm features for algorithm selection. The algorithm features are extracted by an LLM and combined with the problem instance features via cosine similarity. In their experiments, the authors find their method to outperform the baseline methods and in ablation studies using algorithm features is found to play the most important role for performance and training time.

**Strengths:**

- Interesting and novel approach based on LLMs.
- Strong performance in the empirical study compared to the considered baselines.

**Weaknesses:**

- Related literature is not surveyed very well and important parts are not discussed/missing. A comparison to state-of-the-art methods is missing, e.g., SATZilla (Xu, Lin, et al. "SATzilla2012: Improved algorithm selection based on cost-sensitive classification models." Proceedings of SAT Challenge (2012): 57-58.) or Run2Survive (Tornede, Alexander, et al. "Run2Survive: A decision-theoretic approach to algorithm selection based on survival analysis." Asian Conference on Machine Learning. PMLR, 2020.)
- I cannot find a public resource of the datasets that are used. Only a reference to another paper is provided but in this paper I could not find any link to a publicly available resource.
- The approach is not evaluated on the standard benchmark for algorithm selection ASLib. Therefore, it is hard to compare the actual performance gain of the AS method over the state-of-the-art and to what extent algorithm features really help here. Note that ASLib already has some scenarios that are enriched by algorithm features. Therefore the claim of pioneering the usage of algorithm features in algorithm selection also needs to be rejected: (Pulatov, Damir, et al. "Opening the Black Box: Automated Software Analysis for Algorithm Selection." International Conference on Automated Machine Learning. PMLR, 2022.) Still the way of doing it via LLMs might be considered novel.
- It is not clear to what extent algorithm features really help for algorithm selection, as the algorithm features are static and thus remain the same for any problem instance. One can easily prove for decision tree models that algorithm features do not provide any advantage at all compared to just using a single categorical feature naming the corresponding algorithm. Therefore, it is questionable whether the performance gain observed in the experiments can really be attributed to information from the algorithm features or to increased capacity in the model.
- Also the empirical study, as far as I can understand, only considers a single domain of algorithms, i.e., MOEAs. The claimed generality of the approach is thus not proved in the paper. Considering the benchmark scenarios of ASLib might help overcome this limitation.
- Speaking about the datasets considered, the number of problem instances is relatively huge. This setup does not really reflect the real-world. Collecting training data of that size for, e.g., Boolean satisfiability problems or traveling salesperson problems, would most likely be intractable. Therefore, the question is whether the approach also works well enough for scenarios where only a couple hundred or maybe two or three thousand problem instances are available.
- The paper does not explicitly tell what algorithm features are computed exactly and how this methodology works in detail. Overall, the paper is missing many details rendering the work **irreproducible**.
- Various unsupported claims in the introduction that algorithm features represent a valuable source (yet to be proven) etc. Algorithm selection as a field is considered to be very recent "in the past few years" but it is a) quite an old problem proposed by Rice in 1976 and b) already a quite explored field (https://larskotthoff.github.io/assurvey/) with published works in the past decades rather than recent years.

**Questions:**

- How can the authors be sure that information contained in the algorithm features really is the source of information and not only increased capacity?
- Would a label indicating the algorithm have a similar effect for the performance? Why should neural networks in contrast to decision trees (that provably do not benefit from algorithm features) be capable of leveraging algorithm features?
- How general is the approach in fact? What about other domains?
- How well does it compare to well known algorithm selection methods?
- How well does it perform for the ASLib (https://www.coseal.net/aslib/) benchmark?

---

### Official Review · Reviewer_rWvh · 2023-10-30

**Soundness:** 2 fair
**Presentation:** 3 good
**Contribution:** 2 fair
**Rating:** 3
**Confidence:** 4

**Summary:**

The paper proposes to use LLMs to extract features of algorithm code and the
problem instances it solves for better algorithm selection. The authors describe
their method and evaluate it empirically.

**Strengths:**

Interesting topic.

**Weaknesses:**

While the paper is interesting, it is not as fundamentally novel as the authors
claim. There is a recent paper that automatically analyzes algorithms and extracts
features for algorithm selection: https://openreview.net/forum?id=HVxcJbTS8lc
While a different approach is used, this paper does not pioneer the integration
of algorithm features into the algorithm selection process.

Further, the empirical evaluation does not use a standard algorithm selection
benchmark like ASlib (https://www.coseal.net/aslib/), which makes it hard to
compare the results to the literature. Accuracy is not a suitable measure to
evaluate algorithm selection approaches because of the cost-sensitive nature of
the problem that accuracy ignores. Again I refer the authors to ASlib, which
presents more suitable measures.

**Questions:**

No questions.

---

### Official Review · Reviewer_qdqu · 2023-11-01

**Soundness:** 3 good
**Presentation:** 3 good
**Contribution:** 3 good
**Rating:** 5
**Confidence:** 2

**Summary:**

In this paper, the authors propose a method for algorithm selection that jointly relies on algorithm and problem features. The algorithm features are extracted using a pre-trained code LLM, and the problem features are extracted using a tree-based structure. The authors compute a similarity-score and use this for algorithm selection (with some supervision via pos/negative sample selection).
The author find that their approach outperforms other algorithm selection strategies, and also has the scope to be a benchmark task for code LLMs.

**Strengths:**

1) The problem itself is interesting, and seems to be a natural extension of algorithm selection space
2) The authors have carefully understood how to decide negative examples based on a theoretical analysis (and based on the fact that similarity scores are used in the identification of algorithms)
3) Results are interesting and show the impact of the authors approach

**Weaknesses:**

1) It is not clear why the similarity score computed and then a linear layer added -- why not use the similarity score itself for identifying the best matching algorithm? This step would reduce the need for negative sample identification
2) Results on additional datasets would be helpful: are the datasets the authors benchmark standard in the space?
3) It is not clear exactly how authors are comparing to prior work -- can authors describe a bit more about related work?

**Questions:**

1) Can authors clarify if the problem representation method (tree-based) has been used in prior work?
2) Why is the similarity score computed and then a linear layer added? Why not use the similarity score itself for identifying the best matching algorithm?
3) Is this problem the authors study related to code generation for algorithms? Are there relevant baselines/works to cover/compare from this space?